# SpatialTree: Branching Out Spatial Intelligence in MLLMs via a Capability Tree

## Abstract

Spatial Intelligence (SI) is rapidly becoming a cornerstone capability for MLLMs, enabling them to seamlessly perceive, reason about, and interact with complex 3D environments — a critical step towards truly embodied AI systems. However, previous works typically focus on a few specific 3D tasks, offering only a fragmented view of MLLMs' spatial abilities. Inspired by cognitive science studys, we propose SpatialTree, a hierarchical taxonomy that organizes SI into a capability tree—from low level perception (L1), mental mapping (L2), mental simulation (L3), to high level agentic competence (L4). Building on this taxonomy, we introduce the first capability-centric benchmark that thoroughly evaluates the spatial abilities of MLLMs. Moreover, extensive experiments are conducted to investigate the compositional nature of spatial abilities, examining the dependencies among the abilities and identifying the atomic abilities that exert the greatest influence on others. Furthermore, we introduce SpatialEngine, an extensible framework that integrates 3D vision perception models with MLLMs into a progressive annotator, enabling comprehensive data annotation across the entire tree.

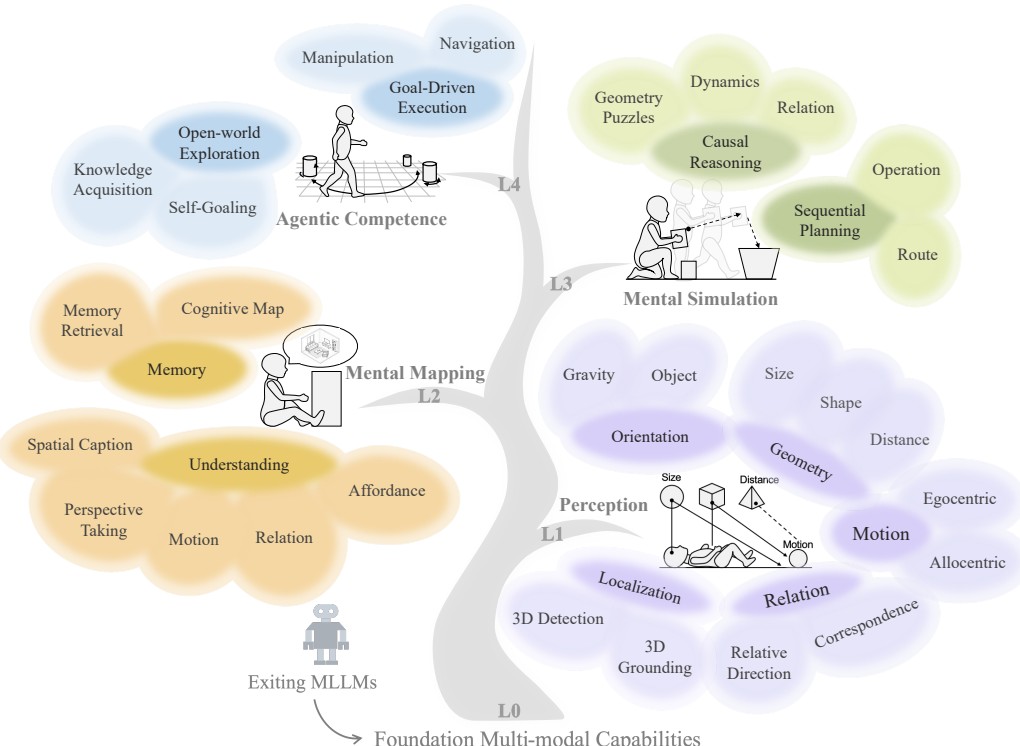

Figure 1: **SpatialTree.** Inspired by cognitive science, our proposed SpatialTree organizes spatial intelligence into a four-layer hierarchy (L1-L4). Rooted in foundational multi-modal capabilities (L0), the tree progressively branches from Basic perception (L1) to agentic competence (L4).

# 1 INTRODUCTION

Developing Spatial Intelligence (SI) Sytems to perceive, reason, and interact within the physical world is a long-standing challenge across cognitive science (Tolman, 1948; Shepard & Metzler, 1988; Newcombe & Huttenlocher, 2000), symbolic AI (Kuipers, 1978; 2000; Harnad, 1990), and robotics (Durrant-Whyte & Bailey, 2006; Thrun, 2002). However, progress has historically been limited by the lack of a unified model capable of integrating perception, reasoning, and action. The emergence of Multimodal Large Language Models (MLLMs), with their powerful vision-language understanding and reasoning capabilities, has opened new opportunities for advancing SI.

Recent research on spatial intelligence (SI) in MLLMs has largely followed a task-centric trajectory. Early works focused on simple spatial tasks in single images (Ma et al., 2024; Wang et al., 2024a; Fu et al., 2024), such as relative object positioning and size estimation. Later studies expanded these tasks to 3D grounding, detection, and captioning from point clouds (Zhu et al., 2024; Hong et al., 2023; Xu et al., 2024). With multi-view and video-capable VLMs, benchmarks quickly diversified (Yang et al., 2025a; Wang et al., 2025d;c; Gholami et al., 2025; Yang et al., 2025c; Jia et al., 2025), covering a wide array of tasks from spatiotemporal reasoning to egocentric and dynamic object understanding.

However, this prevailing task-centric approach, while foundational, has naturally led to a landscape of benchmarks that often focus on specific, sometimes overlapping, spatial tasks. This fragmentation makes it challenging to gain a holistic view of an MLLM's overall spatial intelligence or to understand the inherent dependencies between these skills. This motivates us to ask:

> Can we move beyond fragmented, task-centric benchmarks to uncover a compact
> set of atomic capabilities that capture spatial intelligence and its dependencies?

Inspired by Piaget's theory in cognitive science Piaget (2013), we advocate a *capability-centric* paradigm for spatial intelligence (SI). We further decompose SI into a multi-level capability tree (Fig. 1). Based on this taxonomy, we construct the first comprehensive benchmark for SI in MLLMs, offering comprehensive ability coverage and a diverse set of evaluation metrics beyond simple multiple-choice tests used in prior works. We also develop a Spatial Engine, an extendable annotation framework. It integrates multiple perception models to generate annotations for each capability layer. At the highest level (L4), we propose a spatial action mapping which converts continuous actions into discrete, high-level motion primitives, providing MLLMs with an executable action space for agentic tasks. We leverage the proposed Spatial Engine to generate diverse annotation data from public datasets covering video games, robot manipulation, and human-object interactions, with details provided in Sec. 3.1. To cover the lower levels (L1–L3), we extract relevant portions from multiple public datasets and benchmarks (Yang et al., 2025c;a; Jia et al., 2025; Lin et al., 2025; Wang et al., 2025c; 2024a; Zhu et al., 2024; Yin et al., 2025; Xu et al., 2025; Liu et al., 2025), reorganizing them onto our capability tree. We further enrich questions and evaluation protocols to improve coverage and cross-layer overlap. To address missing capabilities, we introduce **SpatialPlus**, generated by **SpatialEngine**, encompassing Orientation (L1), Memory Retrieval (L2), Relational Reasoning (L3), and Agentic Competence (L4). All resulting data and annotations are systematically organized and re-weighted within the SpatialTree benchmark to ensure balanced evaluation across layers.

Evaluation on SpatialTree-Bench reveals a clear hierarchical structure in spatial intelligence: low-level abilities (L1–L2) are largely independent, whereas higher-level abilities (L3–L4) exhibit strong interdependencies, reflecting their compositional nature. Furthermore, we observe that certain foundational abilities—Geo.Size (L1), Geo.Dist (L1), and Relat.Corr (L2)—as well as higher-level reasoning skills (L3) correlate strongly with agentic competence (L4). To systematically validate these relationships, we design an *atomic prompting* protocol: for L4 navigation tasks, we provide MLLMs with additional prompts encoding relevant L1, L2, and L3 signals. By this, we find lower level information could significantly improves performance on higher 3D agentic tasks, yielding clear gains (e.g., w/ Corres: 12.1%, w/ Depth: 22.1%, w/ Size: 5.1%).

In summary, our work makes the following key contributions:

- **Propose** a capability-centric paradigm for spatial intelligence, offering a systematic and interpretable framework beyond task-centric benchmarks.

- **Construct** the first comprehensive benchmark for spatial intelligence in MLLMs, covering multiple ability layers with diverse evaluation metrics.

- **Develop** a Spatial Engine and a spatial action mapping to generate annotations and enable MLLMs to perform interactive tasks.

- **Validate** the hierarchical classification and inter-level dependencies through experiments, demonstrating that high-level abilities benefit from lower-level information.

## 2 RELATED WORK

**Spatial Cognitive Modeling.** Understanding spatial cognition has long been a central goal in cognitive science and AI. A common insight from classical theories is that spatial abilities are hierarchical, ranging from basic perception and sensorimotor interactions to higher-level reasoning and planning. Piaget Piaget (2013) highlighted the developmental progression of such abilities, Tolman Tolman (1948) introduced the idea of cognitive maps to represent environments for flexible navigation, and Kuipers Kuipers (1978; 2000) formalized a hierarchical spatial representation linking local perception to global knowledge. More recent symbolic and neural approaches Shepard & Metzler (1988); Newcombe & Huttenlocher (2000) extend these insights to computational models of spatial representation, memory, and reasoning. These studies collectively motivate our *SpatialTree*, which organizes spatial intelligence into multi-level capabilities, bridging classical theory with systematic computational evaluation.

**Multi-modal Large Language Models.** The success of GPT-3 Brown et al. (2020) and GPT-3.5 OpenAI (2023a) demonstrated the potential of large language models for complex linguistic understanding and reasoning. GPT-4V OpenAI (2023b) extends GPT-4 Achiam et al. (2023) with visual inputs, enabling single-image understanding and basic spatial reasoning. Open-sourced models such as LLaVA Liu et al. (2023) and QwenVL Bai et al. (2023) gradually added multi-image and video capabilities, supporting spatiotemporal reasoning. Reasoning-augmented LLMs, pioneered by OpenAI O1 Jaech et al. (2024) and DeepSeek-R1 Guo et al. (2025a), integrate chain-of-thought and reinforcement learning to enhance high-level inference. Building on these advances, GPT-4O Hurst et al. (2024) and Gemini 2.5 Comanici et al. (2025) combine perception and reasoning to support complex, agentic decision-making. Collectively, these milestones progressively enable hierarchical spatial intelligence in MLLMs, motivating structured benchmarks and evaluation frameworks across low-level perception, intermediate reasoning, and high-level agentic competence.

**Benchmarks for Spatial Intelligence in MLLMs.** Benchmarks for spatial abilities in MLLMs have evolved alongside the models themselves. Early efforts, such as BLINK Fu et al. (2024), SpatialEval Wang et al. (2024a), and 3DSR-Bench Ma et al. (2024), focused on evaluating spatial understanding tasks in single images, including distance estimation, relational question answering, and spatial captions. As MLLMs increasingly support multi-frame and video inputs, benchmarks such as VSI-Bench Yang et al. (2025a) and MMSI-Bench Yang et al. (2025c) have emerged to evaluate spatial reasoning across multiple views and dynamic scenes. To further enrich task diversity and coverage, Omnispatial Jia et al. (2025), SITE Wang et al. (2025d), and IR3D-Bench Liu et al. (2025) extend benchmarks to geometry puzzles, dynamic reasoning, and inverse rendering tasks. Built upon prior efforts, our SpatialTree benchmark systematically organizes spatial abilities into a hierarchical framework, providing the first thorough evaluation across different capabilities.

## 3 SPATIALTREE: OUR FRAMEWORK FOR SPATIAL INTELLIGENCE

In this section, we present **SpatialTree**, a top-down hierarchical decomposition of spatial capabilities into four levels, from high-level agentic competence (L4) to foundational perception (L1). Different samples for different level of capabilities are shown in Figure 2.

### 3.1 AGENTIC COMPETENCE

We begin from the ultimate objective of a *Spatial AI Agent* — an MLLM-driven system that integrates multi-modal observations, updates its memory, and selects actions to interact with the 3D world in

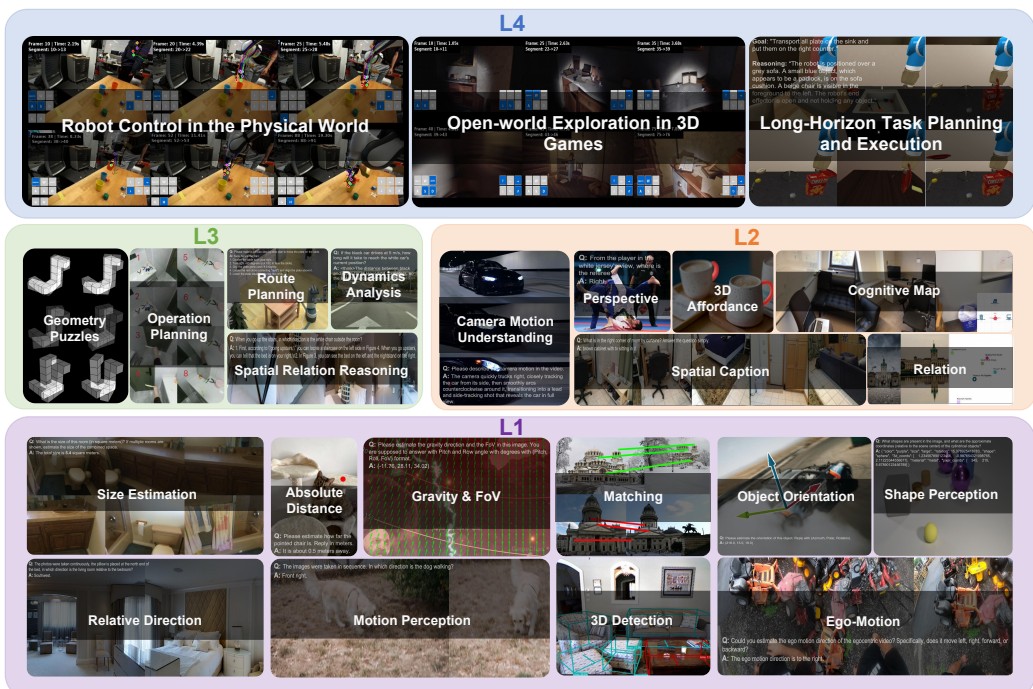

Figure 2: **A gallery of representative tasks.** Leveraging our capability tree, we've built a thorough benchmark covering diverse spatially relevant tasks in all aspects.

an intuitive manner. Formally, the agent performs sequential decision-making by modeling:

$$(S_t, A_t, M_t) \sim P_\theta\Big( \cdot \mid O_t, H_{t-1} \Big), \quad H_{t-1} = \{(O_0, A_0, M_0), \ldots, (O_{t-1}, A_{t-1}, M_{t-1})\} \quad (1)$$

where $O_t \in O$ is the current multi-modal observation, $S_t \in \mathcal{S}$ the internal latent state (e.g., goal, plan, or belief), $A_t \in \mathcal{A}$ the chosen action, and $M_t \in \mathcal{M}$ the updated memory representation. MLLMs are expected to output interactive actions executable across 3D environments and embodiments, such as games, simulators, and the physical world. Unlike Vision-Language Action Models (VLAs) decording the low-level control signals in robotics (Intelligence et al., 2025), MLLMs take the language as the only interface to link with environments like GUI Agents (Qin et al., 2025).

**Spatial Action Mapping.** In the context of spatial agents, navigation and manipulation represent the most common forms of interaction within 3D environments. We address each with a distinct action space design. For navigation, we conceptualize agent movement as a series of camera motion controls (referring to recent video world models (Ball et al., 2025; Mao et al., 2025a)). To enable precise and intuitive control, we decompose complex camera movements into a set of fundamental motion primitives inspired by established cinematography techniques. This approach allows us to translate high-level language instructions (e.g., "move to the left," "look up") into a structured, low-level action space. The six fundamental primitives, their corresponding cinematic terms, degrees of freedom (DoF), and parameterization are detailed in Table 1.

Formally, the camera trajectories are defined with a series of Camera-to-World (C2W) transformation matrices $\mathcal{T}_{\text{motion}} = \{\mathbf{T}|_0^i, i = 0, 1, \ldots, t\}$, while the camera transformation at each moment is $\mathcal{T}_{i \rightarrow i+1} = \mathbf{T}_{i+1}\mathbf{T}_i^{-1}, i = 0, 1, \ldots, t-1$. Then the continuous camera transformation can be decomposed into different components corresponding to each motion primitive, and discretized into the navigation action $A_{\text{nav}}$ using a speed threshold:

$$\mathbf{A}_i^{\text{nav}} = \mathbf{T}_{i \rightarrow i+1} = \{\Delta \mathbf{R}, \Delta \mathbf{t}\} \approx \{t_i \cdot v_i, \ t_k \cdot \omega_k \mid i, k \in \{x, y, z\}, \ t_i, t_k \in \mathbb{Z}_{\geq 0}\}, \quad (2)$$

where $\Delta \mathbf{R} = (\Delta R_x, \Delta R_y, \Delta R_z)$ represents the rotation components obtained via Euler decomposition, $\Delta \mathbf{t} = (\Delta t_x, \Delta t_y, \Delta t_z)$ denotes the translation components along the $x$, $y$, and $z$ axes, and $t_i, t_k$ are discrete integers ranging from 0 up to the video frame rate (FPS). For manipulation, we focus on

Table 1: **Spatial Action Mapping.** This table defines a standardized interface that maps continuous 6-DoF motions and discrete control signals into action primitives with unified parameterization, enabling MLLMs to plan and execute embodied behaviors for agentic competence evaluation.

| Primitive | Primitive Term | Category | Description | Action Mapping | Param. | Threshold |
|---|---|---|---|---|---|---|
| $P_{\text{truck}}$ | Truck | Translation | Move camera left/right (X-axis) | $A/D$ | $v_x$ | $\pm 0.01$ m/s |
| $P_{\text{dolly}}$ | Dolly | Translation | Move camera forward/backward (Z-axis) | $W/S$ | $v_z$ | $\pm 0.01$ m/s |
| $P_{\text{pedestal}}$ | Pedestal | Translation | Move camera up/down (Y-axis) | $Q/E$ | $v_y$ | $\pm 0.01$ m/s |
| $P_{\text{pan}}$ | Pan | Rotation | Turn camera left/right (yaw) | $\leftarrow/\rightarrow$ | $\omega_y$ | $\pm 0.5°$/s |
| $P_{\text{tilt}}$ | Tilt | Rotation | Tilt camera up/down (pitch) | $\uparrow/\downarrow$ | $\omega_x$ | $\pm 0.5°$/s |
| $P_{\text{roll}}$ | Roll | Rotation | Roll camera CW/CCW (roll) | $Z/X$ | $\omega_z$ | $\pm 0.5°$/s |
| $O_{\text{gripper}}$ | Gripper | Gripper Control | Open or close the gripper | $G/H$ | State $\in \{0, 1\}$ | N/A |
| $O_{\text{push/pull}}$ | Push/Pull | Gesture | Push or pull object along forward axis | $P/L$ | Dir. $\in \{-1, +1\}$ | N/A |
| $O_{\text{grab}}$ | Grab | Gesture | Grab or release object | None | State $\in \{0, 1\}$ | $G/H$ |

two representative scenarios to simplify the problem and enable controlled evaluation: human-hand manipulation and robotic gripper manipulation. For the gripper setting, we include gripper open/close actions along with wrist-level 6-DoF motion. For the human-hand setting, we define a small set of intuitive gesture primitives (i.e., push, pull, grab) seen in Table. 1 that capture essential interaction patterns. These manually defined mappings create a unified yet tractable action space for analyzing MLLMs' planning and manipulation competence.

Building on the proposed spatial action mapping, we curate annotated data from diverse sources, including human-hand manipulation videos, navigation video games, robotic arm manipulation datasets, and simulation environments. This unified dataset enables us to evaluate whether MLLMs can accurately plan and execute actions in the defined metric action space. Further implementation details are provided in Sec. 4 and in the experimental section.

### 3.2 MENTAL SIMULATION

Reasoning and planning prior to action execution are essential components of Multimodal Large Language Models (MLLMs), aligning naturally with the Chain-of-Thought paradigm in language model reasoning. In spatial cognitive science, this process is commonly referred to as mental simulation. We further decompose mental simulation into two core components: causal reasoning and sequential planning.

**Causal Reasoning** allows MLLMs to model spatial interactions, physical dynamics, and entity relationships within a simulated mental space. It includes reasoning about object geometry (e.g., how shapes interlock in spatial puzzles), predicting motion under kinematic constraints (e.g., how an object traverses a path), and analyzing semantic–spatial relations (e.g., object A is left of object B). By mentally simulating cause–effect chains in spatial scenarios, MLLMs establish the logical substrate for subsequent planning.

**Sequential Planning** converts causal insights into coherent, goal-directed action plans expressed in language. It entails designing high-level, step-by-step strategies (e.g., "first move toward the door, then turn right, and finally interact with the handle") and generating abstract routes that respect spatial logic (e.g., "go around the table to reach the sofa"). By chaining linguistic action primitives, MLLMs produce strategic plans that ensure the conceptual sequence aligns with the overarching goal before any low-level execution.

### 3.3 MENTAL MAPPING

Level 3's advanced mental simulation requires a coherent internal world model, a foundation provided by Level 2's mental mapping. This process constructs and maintains a dynamic 3D representation of

the environment by relying on two essential facets. The first is spatial understanding: the ability to interpret the immediate scene. This includes recognizing objects and their affordances, mapping the spatial relations between them, and understanding the scene from various perspectives (perspective taking). In essence, it's about making sense of what is currently perceived. The second facet is memory. It allows the agent to retain this understanding over time, retrieving past observations to build a cognitive map that extends far beyond the current field of view. This creates a persistent and comprehensive mental model of the world. Ultimately, these two facets—understanding the present and remembering the past—organize and integrate the foundational perceptual data from L1, paving the way for L3's predictive simulations.

### 3.4 PERCEPTION

Perception forms the foundation for high-level spatial reasoning. We categorize L1 Perception into five core aspects: **Orientation**: Captures spatial alignment, crucial for understanding the agent's pose and maintaining balance. Key sub-tasks are *Gravity* (estimating pitch and roll to determine "up"/"down") and *Object Orientation* (recognizing object poses), supporting scene reconstruction and manipulation. **Geometry**: Involves spatial form, size, and metric relationships. Sub-tasks include *Size*, *Shape*, and *Distance*, enabling reasoning about object properties and facilitating navigation and grasping. **Motion**: Encodes spatial dynamics over time. Sub-tasks are *Egocentric Motion* (self-motion estimation) and *Allocentric Motion* (tracking object or scene changes), critical for predicting future states and planning actions. **Relation**: Concerns spatial relationships between entities. Sub-tasks include *Correspondence* (matching entities across views) and *Relative Direction* (e.g., left of, in front of), supporting object tracking, path planning, and interaction reasoning. **Localization**: Anchors perception within 3D space. Sub-tasks include *3D Detection* (identifying object extents) and *3D Grounding* (associating observations with coordinates), enabling scene reconstruction, navigation, and embodied reasoning.

## 4 SPATIAL ENGINE: OUR DATA ANNOTATOR PIPELINE

We propose an extensible data engine designed to generate annotations for every layer of the SpatialTree. Our approach begins with a diverse set of low-level 3D perception models, each tailored to a specific task, including metric depth estimation (Wang et al., 2025b; Yang et al., 2024), orientation estimation (Wang et al., 2024b), gravity estimation (Veicht et al., 2024), correspondence matching (Leroy et al., 2024; Xiangli et al., 2025), 3D localization (Mao et al., 2025b), 3D point tracking (Xiao et al., 2024; 2025), and camera pose estimation (Wang et al., 2025a;e). Nevertheless, all these comprehensive 3D perception models can be seamlessly encompassed within our taxonomy of five perception abilities.

**Data Annotation Framework.** As shown in Figure 3, our pipeline encapsulates three hierarchical entities: models, pipelines, and workflows. The lowest level comprises the perception models described above, along with advanced MLLMs for semantic captioning. Building upon these models, we construct several reusable pipelines that serve as atomic components for higher-level workflows. Specifically, we implement 12 pipelines, including metric 3D reconstruction, 3D orientation alignment, 3D point tracking, and affordance pointing. Each pipeline processes raw sensory data, such as RGB images or 3D point clouds, and produces intermediate outputs that are further integrated by the workflows. Based on these pipelines, we assemble 24 workflows, each targeting a specific perception ability or a combination of abilities, to generate comprehensive annotations for our SpatialTree. These reusable pipelines not only streamline the annotation process but also facilitate future extension to new tasks or models. Overall, this hierarchical design ensures modularity, scalability, and a clear separation of responsibilities across models, pipelines, and workflows.

**Data Resources.** As seen in Fig. A, our SpatialTree-Bench is constructed by systematically reorganizing a broad range of recent datasets, including VSI-Bench (Yang et al., 2025a), MMSI-Bench (Yang et al., 2025c), LLaVa3D (Zhu et al., 2024), SpatialEval (Wang et al., 2024a), MindCube (Yin et al., 2025), CameraBench (Lin et al., 2025), Omnispatial (Jia et al., 2025), EmbodiedBench (Yang et al., 2025b), SpatialViz (Wang et al., 2025c), Multi-SPA (Xu et al., 2025), and 3DSR-Bench (Ma et al., 2024). To address their scattered capability coverage and over-reliance on simple multiple-choice questions, we first map each question to our SpatialTree framework. We then enhance the evaluation protocol; for instance, complex reasoning tasks from CameraBench and MMSI-Bench are converted

to a hybrid *multi-option + GPT-4 evaluation* format for a finer-grained assessment. Furthermore, to address the remaining gaps in capability coverage, we introduce our **SpatialPlus** dataset. It is specifically designed to target underrepresented abilities such as Orientation (L1), Shape (L1), and Spatial Caption (L2), with a primary emphasis on the complex tasks of Agentic Competence (L4). To generate this data, we leverage our proprietary **SpatialEngine** to automatically create annotations from a diverse array of video sources, including 3D reconstruction datasets, in-game footage Ju et al. (2024), egocentric manipulation videos (Hoque et al., 2025), and robotics data (Khazatsky et al., 2024). More implementation details are discussed in Sec. 6

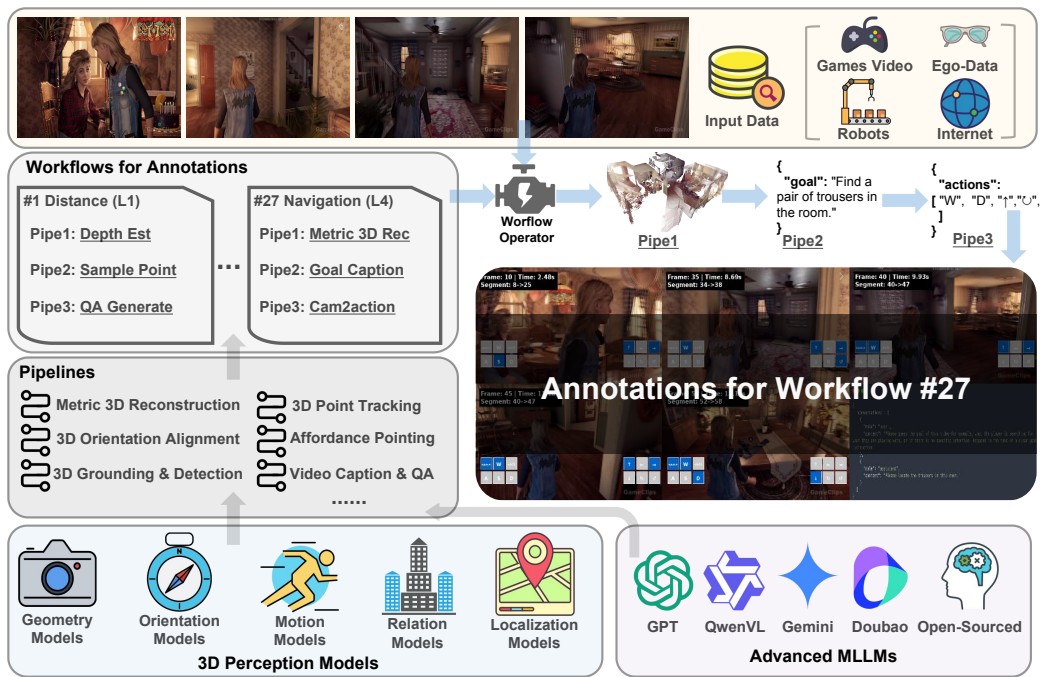

Figure 3: **SpatialTree Data Engine.** A highly modular and scalable framework that decomposes high-level spatial tasks into low-level components, supporting human-in-the-loop supervision.

## 5 EXPERIMENTS

This section presents a comprehensive evaluation of the hierarchical spatial reasoning capabilities of advanced Multimodal Large Language Models (MLLMs) using our *SpatialTree* framework. Our objectives are twofold: 1) to establish a fine-grained capability benchmark for current MLLMs across all levels of the *SpatialTree*, and 2) to analyze the dependencies between foundational spatial skills and their influence on higher-level abilities, such as spatial reasoning.

### 5.1 MODELS AND METRICS

**Benchmarked Models.** We select a diverse set of state-of-the-art MLLMs for our evaluation, ensuring broad coverage of model families, architectures, and functional paradigms. Our selection spans three key dimensions: (1) *reasoning-focused closed-source models* renowned for advanced general reasoning capabilities, including GPT-4o (Hurst et al., 2024), and GPT-5 (OpenAI, 2025); Google's Gemini 2.5 Flash and Gemini 2.5 Pro (Comanici et al., 2025); Anthropic's Claude 3.7 Sonnet (Anthropic, 2025); and GLM-4.5V Hong et al. (2025), SeedVL1.5 Guo et al. (2025b); (2) *non-reasoning models* such as Gemini-2.5-Pro-Nonthink Comanici et al. (2025), Gemini-2.5-Flash-Nonthinking, and SeedVL1.5-Nonthink (Guo et al., 2025b), which represent specialized paradigms outside traditional reasoning-centric designs; and (3) *open-source models* (e.g., Qwen25VL (Bai et al., 2025) series and Kimi-VL (Team et al., 2025)) that reflect the cutting edge of community-driven research. This deliberate diversity allows us to compare performance across reasoning vs. non-reasoning paradigms, closed vs. open-source ecosystems, and varying scales (from 32B to 72B

| Methods | Rank | Avg. | L1 Perception | | | | | L2 Mental Mapping | | L3 Mental Simulation | | L4 Agentic Competence | |
| --- | --- | --- | --- | --- | --- | --- | --- | --- | --- | --- | --- | --- | --- |
| | | | Geom. | Motion | Rel. | Local. | Orient. | Underst. | Memory | Caus. Reas. | Seq. Plan. | Goal Exec. | Open Expl. |
| *Proprietary Models* | | | | | | | | | | | | | |
| GPT-4o | 5 | 44.2 | 43.4 | 30.0 | 59.7 | 73.3 | 41.2 | 61.7 | 41.1 | 37.2 | 53.2 | 19.0 | 22.0 |
| GPT5 | 4 | 46.7 | 44.5 | 34.0 | 58.4 | 77.9 | 36.1 | 60.6 | 55.3 | 38.6 | 52.1 | 23.6 | 19.4 |
| Gemini2.5 Flash NT | 6 | 44.1 | 45.3 | 28.6 | 60.5 | 70.6 | 45.1 | 56.6 | 50.7 | 34.2 | 50.3 | 16.7 | 21.3 |
| Gemini2.5 Pro NT | 3 | 46.9 | 51.4 | 27.9 | 64.2 | 75.1 | 45.4 | 62.8 | 49.8 | 38.6 | 52.9 | 21.8 | 18.3 |
| claude3.7-sonnet NT | 9 | 43.4 | 39.2 | 26.0 | 61.6 | 66.2 | 40.3 | 58.1 | 45.6 | 38.0 | 53.0 | 21.1 | 21.4 |
| SeedVL1.5 NT | 12 | 38.9 | 35.8 | 30.7 | 63.4 | 71.2 | 39.0 | 61.0 | 27.6 | 37.5 | 53.6 | 10.9 | 6.6 |
| *Thinking Models* | | | | | | | | | | | | | |
| SeedVL1.5-Thinking | 7 | 43.5 | 48.0 | 29.3 | 62.6 | 76.4 | 42.1 | 62.4 | 42.0 | 34.3 | 48.2 | 16.3 | 14.7 |
| GLM4.5V | 10 | 42.3 | 48.4 | 25.9 | 67.0 | 71.7 | 42.9 | 54.0 | 42.9 | 37.4 | 52.1 | 17.0 | 9.3 |
| Gemini2.5-Pro | 1 | 50.9 | 53.9 | 33.9 | 64.6 | 77.2 | 45.8 | 62.9 | 60.7 | 44.0 | 55.8 | 24.9 | 24.7 |
| Gemini2.5-Flash | 2 | 47.8 | 42.9 | 25.9 | 62.8 | 75.5 | 42.2 | 59.8 | 60.5 | 36.6 | 53.7 | 22.7 | 25.7 |
| claude3.7-sonnet | 8 | 43.4 | 41.1 | 30.6 | 66.6 | 66.5 | 27.8 | 58.2 | 37.4 | 40.1 | 59.1 | 24.7 | 24.5 |
| *Open-source Models* | | | | | | | | | | | | | |
| Qwen2.5VL-3B | 14 | 32.0 | 29.0 | 29.3 | 38.3 | 53.4 | 33.4 | 43.7 | 28.0 | 26.7 | 41.5 | 18.3 | 11.7 |
| Qwen2.5VL-7B | 16 | 29.0 | 28.2 | 31.2 | 36.2 | 52.4 | 30.3 | 43.0 | 18.6 | 26.6 | 34.5 | 14.3 | 11.8 |
| Qwen2.5VL-32B | 15 | 31.5 | 33.5 | 29.3 | 39.6 | 58.7 | 35.0 | 41.7 | 16.1 | 27.0 | 41.7 | 20.2 | 14.1 |
| Qwen2.5VL-72B | 11 | 41.3 | 38.5 | 22.7 | 59.3 | 66.0 | 35.9 | 59.0 | 36.4 | 32.6 | 53.0 | 23.8 | 20.1 |
| Kimi-VL-A3B-Instruct | 13 | 32.5 | 30.7 | 23.3 | 39.3 | 58.0 | 33.3 | 49.4 | 31.1 | 26.6 | 35.0 | 16.2 | 7.8 |

Table 2: **Our-Bench.** Dark gray indicates the best result among all models and light gray indicates the best result among open-source models. NT denotes the non-thinking model. **Avg** is aggregated by our weighted strategy seen in Sec. 6.

parameters), providing a holistic view of the current MLLM landscape. A complete list of evaluated models is provided in Table 2.

**Evaluation Metrics.** Our evaluation employs a multi-faceted set of metrics tailored to the specific abilities at each level of the SpatialTree. For perception and understanding tasks (L1-L2), we primarily use accuracy-based metrics, such as classification accuracy for object recognition, Mean Squared Error (MSE) for distance estimation, and angular difference for orientation tasks. For higher-level reasoning and planning tasks (L3-L4), we measure task success rates. In the case of agentic tasks (L4), we further analyze the quality of generated actions using metrics like positional error (L2 distance) and orientation error (angular difference) against ground-truth trajectories.

## 5.2 Performance on SpatialTree-Bench

We first present the overall performance of all benchmarked models on our proposed SpatialTree-Bench, with detailed results summarized in Table 2. In our benchmark, the reasoning models achieve clear improvement than their non-thinking version, e.g. Gemini2.5-Pro (53.9) v.s. Gemini2.5-Pro-NT (51.4).

## 5.3 Analysis of Ability Dependencies

To explore the structure of spatial intelligence in MLLMs, we analyze the dependencies among fine-grained sub-abilities using the Pearson correlation coefficient. A high positive correlation indicates that models performing well on one ability tend to perform well on the other. Fig. 4 shows a heatmap of these correlations across all models.

The heatmap suggests a compositional nature in spatial abilities: higher-level capabilities (L3 and L4) exhibit stronger correlation as shown

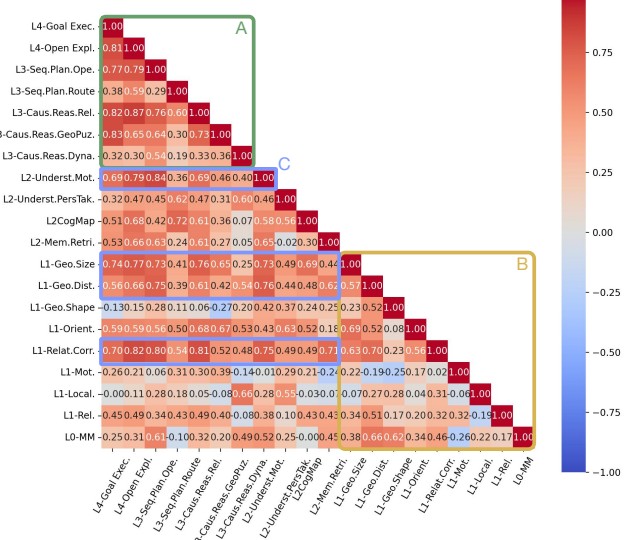

Figure 4: **Inter-Capability Dependencies via Pearson Correlation.** (A) Correlation matrix among higher-level capabilities (L3 and L4); (B) Correlation matrix among foundational L1 capabilities; (C) Salient low-level abilities influencing higher-level tasks.

in region A. This reflects that complex tasks, such as route planning and causal reasoning, depend on overlapping foundational sub-skills. As a result, performance in one high-level ability often predicts

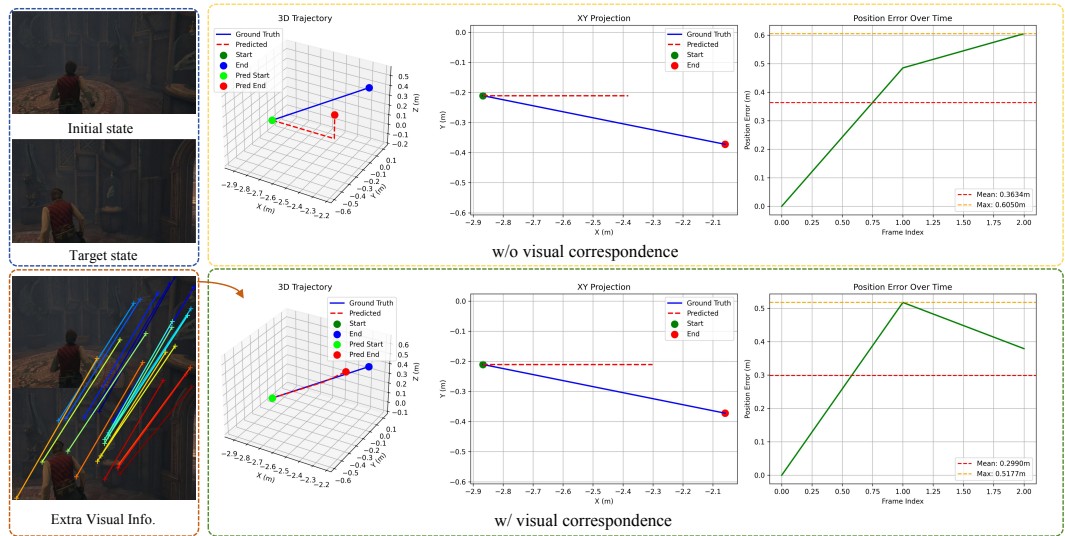

Figure 5: **Correspondence Prompting for Navigation.** The correspondence prompt guides Gemini2.5-pro to navigate and move more accurately within 3D environments.

performance in others. At the lowest level (L1), atomic abilities exhibit weak correlations, indicating that they are largely independent. Foundational skills such as shape perception, distance estimation, and relative direction capture distinct aspects of spatial perception. This orthogonality provides a diverse and comprehensive perceptual foundation for the model.

Finally, we identify a set of low-level critical abilities that act as prerequisites for a wide range of higher-level competencies. For example, strong performance in geometric perception tasks, particularly distance estimation (*L1-Geo.Dist*) and size estimation (*L1-Geo.Size*), shows a strong positive correlation with many advanced abilities, including open exploration (*L4-Open.Expl.*), and causal reasoning (*L3-Seq.Plan.Ope, L3-Caus.Reas.Rel*). This indicates that a model's ability to perceive fundamental geometric properties is a cornerstone upon which more abstract spatial reasoning is constructed. These findings strongly support our hypothesis that a core set of atomic abilities forms the basis for the emergence of broader spatial intelligence in MLLM.

### 5.4 Atomic Prompting for Hierarchical Scaffolding

To investigate the influence of low-level perceptual aids on high-level agentic reasoning, we conducted a controlled experiment on the L4 agentic navigation task (Seen in Fig. 5). The experimental design aimed to isolate the effect of supplementary low-level information while keeping the primary goal and basic visual information consistent across conditions. Specifically, in the baseline condition, the model was provided with visual observations of the initial and final states, alongside a defined 6-Degrees-of-Freedom (6DoF) action space, and was tasked with generating a sequence of actions to connect the two states. In the experimental condition, we augmented the input with explicit visual correspondence figures to provide additional low-level guidance. We evaluated the performance in both settings using our L4 agentic evaluation metric. The results revealed a significant performance uplift: supplying the explicit visual cues improved the model's score by a notable 12%. This finding suggests that even for high-level planning tasks, grounding the reasoning process of MLLMs with explicit, low-level visual information can substantially enhance their performance in complex spatial navigation scenarios.

## 6 Conclusion and Discussion

We propose the first capability-centric Spatial Intelligence framework, SpatialTree, organizing spatial capabilities into four hierarchical layers. Our experiments reveal the compositional structure of these abilities, showing how foundational skills support higher-level performance. Leveraging the scalability of SpatialTree and SpatialEngine, we can systematically generate tasks and annotations guided by the capability hierarchy, providing a framework to enhance pre-training and SFT, and to accelerate the development of next-generation embodied MLLMs.

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

# APPENDIX

## CONTENTS

Figure A: **Construction of SpatialTree-Bench.** We build our benchmark by reorganizing various existing datasets and mapping them to our capability tree, where **SpatialPlus**, a complementary dataset are introduced to ensure the capability coverage.

## A  Visualization of Data Sources

How different datasets contribute to our SpatialTree evaluation is shown in Fig. A.

## B  Evaluation Metrics Details

**Multi-Option QAs.**  For multi-option question answering, each model is evaluated on its ability to select the correct option from a predefined set. We measure accuracy by comparing the predicted choice against the ground-truth answer. This paradigm captures a model's understanding of spatial relations, object properties, and causal dynamics within a scene, corresponding to the low- and mid-level capabilities in the SpatialTree (L1–L3).

**Numeric QAs.**  Numeric QAs require models to predict continuous quantities such as distances, angles, or 3D coordinates. We evaluate performance using relative error metrics, for example:

$$\text{Relative Error} = \frac{|\hat{y} - y|}{|y|},$$

where $\hat{y}$ is the predicted value and $y$ is the ground truth. This metric ensures that predictions are scaled appropriately across different magnitudes and emphasizes precision in spatial reasoning.

**GPT Judge.**  For tasks that are open-ended or involve complex reasoning (e.g., trajectory description, action sequence explanation), we leverage a GPT-based judge to assess correctness. The judge evaluates whether the generated response satisfies the task requirements, optionally scoring partial correctness. This approach allows flexible evaluation beyond rigid numeric or multiple-choice formats, especially for mid- and high-level capabilities in L3–L4.

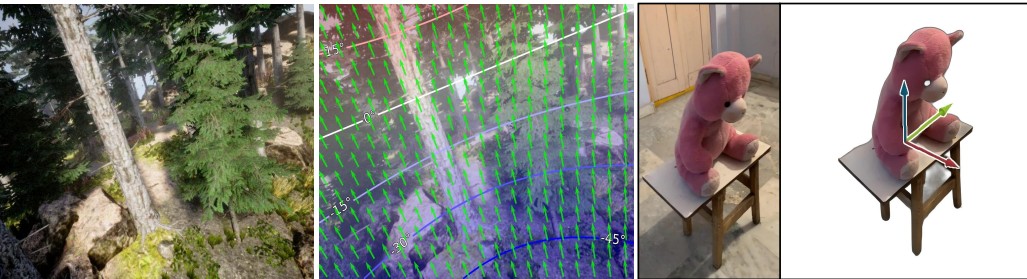

Figure B: **Orientation Annotations.** The left side is the gravity field estimated from GeoCalib Veicht et al. (2024), while the right side is from OrientAnything.

**Agentic Evaluation.** To assess agentic competence, models are deployed in interactive simulated environments, such as those provided by EmbodiedBench (Yang et al., 2025b). We evaluate navigation and manipulation tasks along multiple dimensions: success rate in completing the target goal, relative translation accuracy, and directional alignment. For each action step, a combined metric is computed using relative distance and cosine similarity of movement vectors, producing a normalized score in $[0, 1]$. Aggregating scores over all steps yields a comprehensive measure of an agent's ability to plan and execute actions in long-horizon, embodied tasks.

## C  SPATIALPLUS: COMPLEMENTARY DATA ANNOTATIONS FOR SPATIALTREE

### C.1  ORIENTATIONS (L1)

The Orientation capability, a fundamental yet under-explored area, involves estimating both gravity direction and 3D object orientation. To generate annotations, we leveraged Geocalib Veicht et al. (2024) for gravity vector estimation and OrientAnything Wang et al. (2024b) for object poses. We applied these tools to datasets suited for each task: for gravity, we annotated 500 images sampled from the diverse, drone-captured TartanAir Wang et al. (2020) dataset; for object orientation, we utilized the object-centric Co3dv2 Reizenstein et al. (2021) dataset (Seen in Fig. B). For gravity, the goal is to estimate the camera's orientation relative to the gravity vector, typically represented by the *pitch* and *roll* angles. Formally, let the gravity vector in the world frame be:

$$\mathbf{g}_w = \begin{bmatrix} 0 \\ 0 \\ -1 \end{bmatrix}, \tag{3}$$

and let $\mathbf{R}_{cw} \in SO(3)$ denote the rotation from the world frame to the camera frame. The gravity direction in the camera frame is then:

$$\mathbf{g}_c = \mathbf{R}_{cw}\,\mathbf{g}_w. \tag{4}$$

From $\mathbf{g}_c = [g_x, g_y, g_z]^\top$, the pitch and roll angles can be computed as:

$$\text{pitch} = \arctan 2(-g_x, \sqrt{g_y^2 + g_z^2}), \tag{5}$$

$$\text{roll} = \arctan 2(g_y, g_z). \tag{6}$$

Here, *pitch* measures the forward–backward tilt of the camera, while *roll* measures the sideways tilt. To evaluate an MLLM's proficiency in this task, we require the model to analyze the input image and return these same three parameters in a structured JSON format. An example of our prompt template is shown in Listing 1.

```
1   {
2     "role": "system",
3     "content": "You are a vision model specialized in estimating camera
      ↪  orientation from images.
4     Your task is to infer the gravity direction from the input image by
      ↪  predicting the
5     camera's pitch and roll angles, as well as the vertical field of view
      ↪  (vFOV).
6     Always output your prediction strictly in the following JSON format:
7     {
8       \"pitch\": <float, camera pitch angle in degrees>,
9       \"roll\": <float, camera roll angle in degrees>,
10      \"vfov\": <float, vertical field of view in degrees>
11    }
12    Do not include any additional text or explanation outside of the JSON
      ↪  object."
13  }
```

Listing 1: **Prompt template** for Orientation Estimation.

For evaluation, we move beyond a simple absolute error metric and adopt a probabilistic approach that accounts for the inherent uncertainty of the ground-truth annotations provided by *Geocalib*. For each predicted parameter (pitch, roll, and vFOV), *Geocalib* also outputs an uncertainty value, which we interpret as the standard deviation ($\sigma_{gt}$). We then calculate a normalized similarity score ($S$) for each parameter using a Gaussian kernel, defined as:

$$S(y_{\text{pred}}, y_{\text{gt}}, \sigma_{\text{gt}}) = \exp\left(-\frac{(y_{\text{pred}} - y_{\text{gt}})^2}{2\sigma_{\text{gt}}^2}\right) \tag{7}$$

where $y_{\text{pred}}$ is the MLLM's prediction, $y_{\text{gt}}$ is the ground-truth value from *Geocalib*, and $\sigma_{\text{gt}}$ is its associated uncertainty. This scoring function gracefully penalizes deviations from the ground truth: the score is 1 for a perfect match and decays towards 0 as the error increases. Crucially, a larger uncertainty $\sigma_{\text{gt}}$ in the ground truth leads to a slower decay, making the scoring more lenient when the ground truth itself is less certain. The final score for the task is the average of the individual scores for pitch, roll, and vFOV. For object orientation estimation, most of metrics are similar to gravity, and the evaluation are conducted on Azimuth, Polar and Rotation these three angles.

### C.2 AGENTIC COMPETENCE (L4)

### C.3 GOAL-DRIVEN NAVIGATION

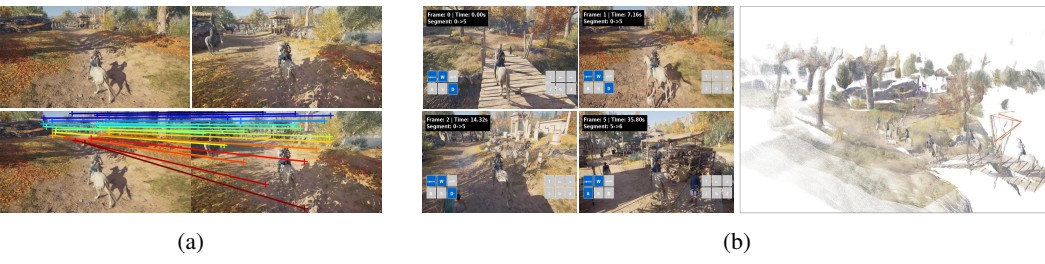

(a)                                                      (b)

Figure C: **Navigation Data Curation.** (a) shows paired images used for evaluation, where MLLMs are expected to move from left to right. (b) illustrates our curation process: reconstructing metric 3D models and camera trajectories, then converting them into actions.

**Goal-driven Navigation.** We leverage our SpatialEngine to get the action annotations as shown in Fig. C. We first extract the metric pose trajectories from the games videos, and convert them into discrete actions with our spatial action mapping, and then we randomly sample several image pairs from the video with the correspondence checking. For evaluation, the goal is a image, and the MLLMs are supposed to control the character to move to the target positions. We use the prompt template as below:

```
{
  "role": "system",
  "content": "Task Details:\n
Analyze Images: Compare the start image <Image 1> and the target image <Image
↪  2> to understand the required translation and rotation for the robot
↪  arm's end-effector.\n
Define Motion: Decompose the movement into 6 steps, each containing one or
↪  more elementary actions.\n
Quantify Actions: For each action, specify an integer step_num that
↪  represents its intensity.\n\n
Coordinate System:\n
Right-hand frame attached to the end-effector: +Z forward, +X right, +Y
↪  downward.\n\n
Action Space:\n
Translation: Dolly In (W), Dolly Out (S), Truck Left (A), Truck Right (D),
↪  Pedestal Up (space), Pedestal Down (shift).\n
Rotation: Pan Left (←), Pan Right (→), Tilt Up (↑), Tilt Down (↓), Roll CW
↪  (Ⓡ), Roll CCW (Ⓛ).\n
Special Action: Stay (STOP).\n\n
Step Size:\n
Translation: 0.019375 m/step. Rotation: 0.4509 rad/step.\n\n
Output Format:\n
Return a single JSON object with keys step_1-step_6. Each step contains:\n
  actions: list of action symbols\n
  step_nums: corresponding integers.\n\n
Example:\n
{
  \"step_1\": {
    \"actions\": [\"W\", \"A\"],
    \"step_nums\": [5, 2]
  },
  \"step_2\": {
    \"actions\": [\"W\", \"↑\"],
    \"step_nums\": [3, 4]
  }
}"
}
```

Listing 2: **Prompt of navigation.**

In this prompt, translation and rotation steps are computed from the actual movement, while capping the number of steps at 10 to prevent overly long action sequences. To evaluate MLLMs, we compute a normalized metric in the range $[0, 1]$ by combining **relative distance** and **directional accuracy**. Specifically, for each step, let $\Delta \mathbf{p}_{\text{pred}}$ and $\Delta \mathbf{p}_{\text{gt}}$ denote the predicted and ground-truth translation vectors, respectively.

The **relative distance score** is defined as:

$$s_d = \max\left(0, 1 - \frac{\|\Delta\mathbf{p}_{\text{pred}} - \Delta\mathbf{p}_{\text{gt}}\|}{\|\Delta\mathbf{p}_{\text{gt}}\|}\right),$$

and the **directional score** is computed by the cosine similarity:

$$s_\theta = \frac{\Delta\mathbf{p}_{\text{pred}} \cdot \Delta\mathbf{p}_{\text{gt}}}{\|\Delta\mathbf{p}_{\text{pred}}\| \, \|\Delta\mathbf{p}_{\text{gt}}\|}.$$

The final step-wise accuracy is then: $s_{\text{step}} = s_d \cdot \max(0, s_\theta)$

which ensures a value in $[0, 1]$, where 1 indicates perfect alignment in both distance and direction. Aggregating $s_{\text{step}}$ across all steps provides a comprehensive measure of the model's precision in executing end-effector motions.

**Goal-driven Manipulation** For the **Goal-Driven Manipulation** capability, we utilize action annotations from the `Droid` Khazatsky et al. (2024) and `EgoDex` Hoque et al. (2025) datasets. This task requires the MLLM to generate a sequence of precise actions to move a robot end-effector or a human hand from a starting state to a target state, both specified by images. The action space for `Droid` encompasses 7-DoF control: 6-DoF for the end-effector's pose (translation and rotation) and a binary state for the gripper (open/close). A similar action space is adapted for `EgoDex`, controlling wrist pose and finger grip. The MLLM is prompted to generate a sequence of continuous action vectors, as shown in the template below:

To evaluate the MLLM's performance, we assess the accuracy of the predicted action sequence against the ground truth. For the translational component of the motion, we reuse the step-wise accuracy metric $s_{\text{step}}$ from the navigation task, which combines relative distance and directional scores. For the rotational component, we compute a normalized score based on the angular difference between the predicted orientation and the ground truth. Let $R_{\text{pred}}$ and $R_{\text{gt}}$ be the predicted and ground-truth rotation matrices for a step. The rotational error angle $\theta_{\text{err}}$ is calculated from the error rotation matrix $R_{\text{err}} = R_{\text{pred}}R_{\text{gt}}^T$:

$$\theta_{\text{err}} = \arccos\left(\frac{\text{Tr}(R_{\text{err}}) - 1}{2}\right).$$

The **rotation score** $s_{\text{rot}}$ is then defined as:

$$s_{\text{rot}} = \max\left(0, 1 - \frac{\theta_{\text{err}}}{\pi}\right),$$

which normalizes the error to a $[0, 1]$ range, where 1 indicates a perfect rotational match. Finally, the **gripper score** $s_{\text{gripper}}$ is a binary accuracy (1 if the predicted state matches the ground truth, 0 otherwise). The final score for each step is a weighted combination of these three metrics, providing a holistic evaluation of the model's ability to perform precise, multi-faceted manipulation tasks.

## D  EMBODIED AGENT EVALUATION WITHIN SIMULATION

EmbodiedBench (Yang et al., 2025b) provides a closed-loop evaluation framework in which MLLMs are deployed within interactive simulators. It includes four primary environments—*EB-ALFRED*, *EB-Habitat*, *EB-Navigation*, and *EB-Manipulation*—supporting long-horizon tasks that require both high-level planning and low-level control. Following the benchmark's evaluation protocol, we assess our models' navigation and manipulation capabilities in these simulated settings.

## E  BENCHMARK METRIC AGGREGATION

To derive a single, comprehensive score for a model's spatial intelligence, we employ a hierarchical aggregation methodology. This approach is designed to reflect the complex, multi-layered nature of spatial cognition, rather than treating all abilities as equally important. The design is principally guided by established theories in cognitive psychology, which posit that spatial intelligence is constructed hierarchically, with fundamental perceptual skills forming the bedrock for more abstract reasoning and planning.

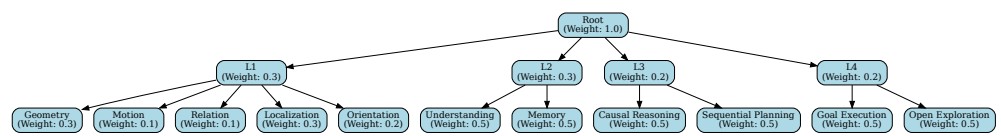

Figure D: An illustration of the hierarchical weighting scheme for metric aggregation with in the *SpatialTree*. Each node represents a capability layer, with the assigned weight used for the bottom-up calculation of the final score. The weighting prioritizes foundational perceptual abilities (L1) as they are prerequisites for higher-level cognitive tasks.

Our aggregation framework is built upon the *SpatialTree* structure. The assignment of weights within this tree is determined by a synthesis of theoretical principles and empirical, data-driven insights:

**Cognitive Hierarchy.** In line with cognitive science literature, our weighting scheme prioritizes foundational capabilities, as shown in Fig. D. The L1 layer, which represents low-level spatial perception, is assigned the largest weight, as these skills are prerequisites for almost all higher-level spatial tasks found in L2 (Mental Mapping) and L3 (Mental Simulation).

**Empirical Dependency from Correlation Analysis.** The theoretical hierarchy is further refined and validated by our empirical findings from the Pearson correlation heatmap (Fig. **??**). The heatmap allows us to identify *atomic abilities* that exhibit strong, widespread correlations with a multitude of other skills. These influential abilities are considered more fundamental to the overall spatial intelligence network and are consequently assigned higher weights within their respective sub-trees. This ensures our metric is not just theoretically sound, but also reflects the actual dependencies observed in model performance.

The final score is calculated via a bottom-up, weighted summation. The performance score for any parent node in the tree is the weighted sum of the scores of its immediate children. This process is recursively applied until the root node is reached, yielding a single, principled score that holistically quantifies the spatial intelligence of a given MLLM.

# F  LLM Usage Declarations

We declare that Large Language Models (LLMs) were used in a limited capacity during the preparation of this manuscript. Specifically, LLMs were employed for grammar checking, word choice refinement, and typo correction. All core technical contributions, experimental design, analysis, and conclusions are entirely our own. The use of LLMs did not influence the scientific methodology, result interpretation, or theoretical contributions of this research.

```
{
  "role": "system",
  "content": "Task Details:\n
Compare the start image <Image 1> and target image <Image 2> to infer the
↪   translation and rotation required for the robot arm's end-effector.\n
Decompose the motion into up to 6 steps, each combining any number of
↪   elementary actions.\n\n
Action Space:\n
We define a 7D action vector per step:\n
[dx, dy, dz, d_roll, d_pitch, d_yaw, gripper_state]\n
- Translation (dx, dy, dz): Displacement in meters along +X, +Y, +Z.\n
- Rotation (d_roll, d_pitch, d_yaw): Rotation in radians about +Z, +X, +Y
↪   respectively.\n
- gripper_state: 0=open, 1=closed.\n\n
Each dx, dy, dz, d_roll, d_pitch, d_yaw is computed from selected actions and
↪   their step_nums:\n
Δq = step_num × unit_step_size  (translation in meters or rotation in
↪   radians)\n\n
Available Actions:\n
W/S: Dolly In/Out (+/-Z)\n
A/D: Truck Left/Right (-/+X)\n
space/shift: Pedestal Up/Down (-/+Y)\n
←/→: Pan Left/Right (± yaw)\n
↑/↓: Tilt Up/Down (± pitch)\n
Ⓡ/Ⓛ: Roll CW/CCW (± roll)\n
STOP: No movement\n\n
Output Format:\n
Return a single JSON object where each step is a key (\"step_1\", \"step_2\",
↪   ...).\n
Each step contains:\n
- actions: a list of action symbols\n
- step_nums: a list of integers specifying intensity (1-10)\n
- gripper: 0 or 1 for gripper state\n\n
Example:\n
{
  \"step_1\": {
    \"actions\": [\"W\", \"A\"],
    \"step_nums\": [5, 2],
    \"gripper\": 0
  },
  \"step_2\": {
    \"actions\": [\"Ⓡ\"],
    \"step_nums\": [3],
    \"gripper\": 1
  }
}"
}
```

Listing 3: **Prompt for Goal-Driven Manipulation with 7D Action Representation.**

