# OpenReview forum: "SpatialTree: Branching Out Spatial Intelligence in MLLMs via a Capability Tree"
_ICLR.cc/2026/Conference — ICLR 2026 Conference Withdrawn Submission_

### Official Review · Reviewer_36Yc · 2025-11-01

**Soundness:** 3
**Presentation:** 3
**Contribution:** 3
**Rating:** 4
**Confidence:** 4

**Summary:**

This paper introduces SpatialTree, a suite of spatial benchmarks organized into a hierarchical taxonomy, referred to as a capability tree. SpatialTree arranges 12 benchmarks into a four-level hierarchy: low-level perception (L1), mental mapping (L2), mental simulation (L3), and high-level agentic competence (L4). To address capabilities not covered by existing benchmarks, the authors developed SpatialPlus. The experiments evaluated mainstream open-source and Proprietary models of various sizes. Results show that lower-level capabilities are relatively independent, while higher-level capabilities exhibit strong interdependencies.

**Strengths:**

- SpatialTree introduces a hierarchical taxonomy that organizes 12 benchmarks into a four-level structure: L4 (robotics tasks), L3 (route planning, etc.), L2 (perspective-taking, cognitive mapping, etc.), and L1 (perception tasks such as metric measurment). Figure A clearly illustrates how existing datasets are reorganized into these four capability levels.
- To address capabilities not covered by existing benchmarks, the authors developed SpatialEngine to synthesize the SpatialPlus dataset.
- Comprehensive experiments were conducted to evaluate mainstream open-source and Proprietary models of various sizes. Particularly, Figure.4 analyzes the relationships between capabilities. Results indicate that lower-level capabilities (L1/L2) are relatively independent, while higher-level capabilities (L3/L4) exhibit strong interdependencies and benefit from low-level information.

**Weaknesses:**

I believe it is important to provide a more thorough discussion of recent works.

**Benchmark Scope**

- Several contemporaneous studies ([1,2,3]) share a similar scope with this work, focusing on collecting or re-annotating existing benchmarks. A detailed comparison of the similarities and differences between these studies and this work is necessary. Additionally, given these works, it may be worth reconsidering the statement in L108 claiming this as “the **first** comprehensive benchmark for spatial intelligence in MLLMs.”
- CoreCognition [4] also conducted task correlation analyses, and SpatialTree’s Figure 4 is notably similar to their analysis. The relationship between low-level and high-level capabilities could further benefit from a comparison with CoreCognition's findings. Moreover, since CoreCognition is also grounded in cognitive science, it would be valuable to discuss the similarities and differences between its taxonomy and that of SpatialTree.

**Experimental Results**

- The observations in Table 2 differ from those in [2,3], where GPT-5 demonstrated strong spatial capabilities. Notably, [2] highlights GPT-5's performance under different thinking modes, while in Table 2, GPT-5 is not classified as a thinking model. The authors may need to clarify the experimental settings for GPT-5.
- Sec. 5.4 (L466-477) shows that augmenting input images with visual correspondence annotations led to a 12% performance improvement for Gemini-2.5-pro in L4 tasks. However, this contrasts with observations in MMSI [5], where its Figure 5 shows that visual prompting yields only slight performance gains. This discrepancy needs further discussion.

[1] SITE: towards Spatial Intelligence Thorough Evaluation

[2] Holistic Evaluation of Multimodal LLMs on Spatial Intelligence

[3] How Far are VLMs from Visual Spatial Intelligence? A Benchmark-Driven Perspective

[4] CoreCognition: Core Knowledge Deficits in Multi-Modal Language Models

[5] MMSI-Bench: A Benchmark for Multi-Image Spatial Intelligence

**Questions:**

I believe this paper needs to include more comparisons with recent benchmarks in terms of scope and experimental results. Details can be found in the weakness section. I am willing to raise my score if my concerns are well addressed.

---

### Official Review · Reviewer_KKTx · 2025-11-01

**Soundness:** 3
**Presentation:** 3
**Contribution:** 3
**Rating:** 6
**Confidence:** 3

**Summary:**

This paper proposes SpatialTree, a hierarchical framework to evaluate spatial intelligence in multimodal large language models (MLLMs).
It organizes spatial abilities into four levels—perception, mental mapping, mental simulation, and agentic competence—and builds the SpatialTree-Bench benchmark with balanced coverage across these layers.
Using SpatialEngine, the authors generate annotations and systematically analyze inter-level dependencies, showing that foundational perceptual skills (e.g., geometry, distance) support higher-level reasoning and navigation.

**Strengths:**

Novel Capability-Centric Framework:
The hierarchical “SpatialTree” design offers a structured and interpretable perspective on spatial intelligence in MLLMs, bridging cognitive science theory with computational evaluation.

Comprehensive Benchmarking:
SpatialTree-Bench integrates diverse datasets and introduces SpatialPlus to fill ability gaps, covering perception to action-level intelligence. The breadth of coverage surpasses prior spatial reasoning benchmarks (e.g., MMSI-Bench, VSI-Bench).

Systematic Analysis:
The work quantifies inter-capability dependencies (Fig. 4) and empirically demonstrates that low-level perception abilities correlate with high-level reasoning and planning. This analysis is both original and insightful.

Practical Tooling Contribution:
SpatialEngine provides a modular annotation and evaluation pipeline that can be reused for future benchmarks or training pipelines for embodied AI.

Strong Empirical Evaluation:
The authors benchmarked over a dozen advanced MLLMs under consistent settings (Table 2), offering valuable insights into model hierarchy and reasoning performance.

**Weaknesses:**

Limited Novelty in Model Design:
The paper’s primary contribution lies in benchmarking and taxonomy design, rather than proposing new MLLM architectures or learning algorithms. The “SpatialEngine” and evaluation setup are valuable but largely engineering extensions.

Benchmark Construction Complexity:
The pipeline integrates many datasets and annotations, but the details of harmonization and standardization (e.g., difficulty balancing, metric normalization) are not deeply elaborated. This may affect reproducibility.

Evaluation Bias:
While broad, the benchmark still relies heavily on simulated and indoor datasets (e.g., Matterport3D, games, robotics footage), limiting ecological diversity. Outdoor or large-scale geospatial reasoning is not addressed.

Limited Validation of Hierarchical Causality:
The paper infers inter-level dependencies through correlations, but does not demonstrate causal relationships (e.g., ablating perception abilities to observe downstream degradation).

Interpretability of “Capability Scores”:
The paper defines multiple overlapping metrics (accuracy, error distance, GPT-judged reasoning), but it is unclear how scores across layers are calibrated or weighted for fair comparison.

**Questions:**

See Weaknesses

---

### Official Review · Reviewer_qkT6 · 2025-11-05

**Soundness:** 1
**Presentation:** 3
**Contribution:** 2
**Rating:** 4
**Confidence:** 3

**Summary:**

This paper presents SpatialTree, a novel capability-centric framework for systematically analyzing, evaluating, and enhancing spatial intelligence in multimodal large language models (MLLMs). To operationalize this framework, the authors develop SpatialEngine, a scalable data generation and annotation engine that supports multi-level task construction across spatial reasoning subtasks. Leveraging these tools, the paper introduces a benchmark encompassing more than 15 spatial subtasks and conducts a hierarchical analysis of inter-level dependencies across prominent MLLMs such as GPT-4V, Gemini, and Qwen-VL. Overall, the paper contributes both a conceptual hierarchy and a comprehensive benchmark, providing a structured lens for diagnosing and potentially improving the spatial reasoning capabilities of multimodal foundation models.

**Strengths:**

[1] The proposed SpatialEngine represents a strong engineering and methodological contribution, offering a scalable and reproducible pipeline for generating multi-modal, multi-level spatial reasoning datasets.
[2] The SpatialTree framework presents a meaningful conceptual advance by organizing spatial reasoning into a hierarchical capability structure. This enables more interpretable diagnostics and may guide future research in compositional generalization, embodied reasoning, and spatial grounding for robotics and vision-language models.

**Weaknesses:**

[1] While the analysis is comprehensive, the paper primarily delivers diagnostic insights rather than methodological advances. To enhance practical impact, the authors are encouraged to complement their analytical framework with concrete improvement strategies, such as targeted training interventions, adaptive curriculum design, or model adaptation techniques informed by the proposed hierarchy.
[2] Many spatial subtasks are synthetically generated via SpatialEngine, which raises concerns about ecological and practical validity. It remains unclear whether the observed performance patterns in these synthetic benchmarks generalize to real-world embodied reasoning tasks, such as 3D navigation or physical manipulation.
[3] The claimed hierarchical dependencies between low-level (L1–L2) and high-level (L3–L4) spatial capabilities are inferred primarily through correlation-based visualization. However, without causal or intervention-based experiments (e.g., selectively perturbing lower-level representations or excluding specific capability subsets), it is difficult to establish causal grounding of the hierarchy. Including such analyses would substantiate the “capability tree” argument and strengthen the paper’s empirical validity.

**Questions:**

[1] Have the authors tested whether improvements in SpatialTree-derived metrics (e.g., L1–L3) correlate with actual performance gains in embodied tasks, such as manipulation or navigation?
[2] Can atomic prompting be formalized into a modular prompting strategy compatible with multi-turn dialogue or interactive reasoning?
[3] While SpatialEngine generates synthetic annotations, how are the correctness and diversity of the tasks validated? Are there human evaluations or inter-rater checks?

---

### Official Review · Reviewer_fppU · 2025-11-12

**Soundness:** 1
**Presentation:** 3
**Contribution:** 2
**Rating:** 2
**Confidence:** 3

**Summary:**

This paper addresses spatial intelligence (SI) in LMs, a critical capability to bridge the gap between LMs and robotics. They introduce a hierarchy of capabilities which they dub “spatialtree” which can inform evaluation of SI in LMs, and build that into a benchmark using SpatialEngine, a framework for annotating various subtasks from the tree on 3D data.

They start from a hypothetical “spatial AI agent” to describe the ideal of SI. They use cinematography concepts like panning tilting and trucking to introduce primitives around moving the camera, which are formalized as transformation matrices. Similar analogies are used to motivate other elements in the SpatialTree hierarchy, such as mental mapping, perception, etc.

Using SpatialEngine they reorganize existing benchmark samples to capture SI. They build these using some undescribed pipelines, into a SpatialBench benchmark, which they use to evaluate a broad set of models.

**Strengths:**

Ambitious effort to taxonomize spatial intelligence.

Novel benchmark that (allegedly) aligns with the proposed schema

Extensive evaluation of free weight and proprietary models. The benchmark isn’t saturated yet, so it should be useful going forward.

**Weaknesses:**

The paper spends a lot of time introducing fancy concepts, for sparse description of methods. A lot of the equations are superfluous, taking up space that would be better spent on concrete examples. In general, too much “what” and not enough “how.”

I have trouble understanding the value of the SpatialTree taxonomy, which is really central to the paper. Lots of time is spent on the “what” and very little on the “how”. An emblematic passage is sections 3.2 and 3.3. They describe concepts that may be important for SI like causal reasoning, sequential planning, and “mental mapping,” but they provide no details on how those actually are evaluated.

How am I supposed to assess how well your benchmark captures “mental simulation” when you haven’t even said which datasets those specific samples have come from? I would really need to see a list of tasks instead of hand wavy “e.g.” phrases here.

What are the data statistics? How many samples in what domains?

(Line 307-311): “We construct several reusable pipelines…” how do you do this? What are these pipelines? What are the expected outputs?

All of these are very basic main text stuff.

Frankly, I do not come away with an understanding of what you concretely did when I read the paper.

**Questions:**

Curious to see your reactions to weaknesses. Please correct any misunderstandings.

---

### Note · Authors · 2025-11-14

I have read and agree with the venue's withdrawal policy on behalf of myself and my co-authors.